# Growth Allocation Shifts in the Invasive *Hydrilla verticillata* Under Interspecific Competition with Native Submerged Macrophytes

**DOI:** 10.3390/plants13243500

**Published:** 2024-12-15

**Authors:** Letícia da Costa, Luíz Alberto Vieira, Thaísa Sala Michelan, Alvaro Herrera Vale, Wagner Antonio Chiba de Castro

**Affiliations:** 1Neotropical Biodiversity Graduate Program, Federal University of Latin American Integration, Foz do Iguaçu 85866-000, PR, Brazil; betaobio43@gmail.com (L.A.V.); alvarovale.h@gmail.com (A.H.V.); wagner.castro@unila.edu.br (W.A.C.d.C.); 2Institute of Biological Sciences, Federal University of Pará, Belém 66075-110, PA, Brazil; thaisamichelan@gmail.com; 3Latin American Institute of Life and Nature Sciences, Federal University of Latin American Integration, Foz do Iguaçu 85866-000, PR, Brazil

**Keywords:** submerged macrophytes, invasive exotics, natives as barriers, competition

## Abstract

Communities with high native species diversity tend to be less susceptible to the establishment of invasive species, especially in studies that test their local impact. This study investigated the impact of competition between native submerged aquatic macrophytes (SAMs) (*Egeria najas* and *Egeria densa*) and the exotic *Hydrilla verticillata*, recognized for its invasive potential in aquatic ecosystems, through a mesocosm experiment conducted over six months. Two treatments were evaluated: the intraspecific competition of *H. verticillata* and an interspecific competition involving all three species. The plants were cultivated under controlled conditions, with the foliar and subterranean biomass being monitored to analyze resource allocation patterns. The results showed that, under interspecific competition, the mean foliar biomass of *H. verticillata* was significantly higher compared to the intraspecific treatment, while the subterranean biomass was reduced in the presence of native species. We conclude that native species alter the biomass allocation pattern of the invader, favoring foliar structures over subterranean ones. Among the mechanisms of invasiveness, differential resource allocation represents an important strategy for the adaptation and competitiveness of invasive species influenced by environmental factors and resource competition. These findings contribute to the understanding of competitive interactions in aquatic ecosystems and have implications for the management and control of invasive species, highlighting the importance of promoting native diversity to mitigate invasibility. Future studies should investigate the impacts of reduced subterranean structures on the persistence and spread of submerged invasive species.

## 1. Introduction

Aquatic macrophytes play a structural role in lotic and lentic environments, promoting habitat heterogeneity and diversity in natural and artificial ecosystems and contributing to ecosystem services [1,2,3,4]. However, their overabundance can cause environmental impacts such as biodiversity loss, eutrophication, and economic and recreational losses, in addition to compromising the very ecosystem services they support [5,6,7].

The dominance of macrophytes is often associated with damming using hydroelectric reservoirs, which alters aquatic ecosystems and facilitates biological invasions [8,9,10]. In lentic environments, favorable conditions can accelerate the growth and spread of exotic species, driving physicochemical and biological changes in the habitat, as well as species composition shifts [11,12,13].

While communities with greater native diversity tend to resist invasive species establishment, the effectiveness of this barrier remains uncertain, varying with environmental changes and species interactions [14,15,16]. Experiments suggest that low native species diversity facilitates the occurrence of exotic species and that exotic species closely related to natives are potentially more invasive [17,18]. When the invader is functionally similar, it may become a naturalized member of the community [19]. For a successful plant invasion, the exotic species must be functionally different from the natives, but not too much [20]. On the other hand, other studies emphasize that both similarities and differences can be determinants of invasive plant success [21,22].

Moreover, there is still uncertainty regarding whether abundance and species richness act as a barrier [14,15], the extent to which the barrier effect is due to abiotic factors [16], or the development patterns of the involved species [23]. For an invasive plant species to establish and spread, it is necessary for the community to allow the exotic species to thrive initially [24]. Over time, coexistence between the exotic species and the native community would increase spatially, causing the biotic resistance of even functionally diverse communities to weaken. This absence of a biodiversity barrier to plant invasion suggests that coexistence between the invader and the native community is more important than exclusion through competition [25].

The coexistence between exotic and native macrophytes, in addition to being influenced by phylogenetic proximity, can also be influenced by phylogenetic and morphological differences. An example is the invasive macrophyte *Hydrilla verticillata* (L.f.) Royle, which, in a specific community in the Rosana Reservoir in southern Brazil, did not face biotic resistance and established alongside species that were less phylogenetically and morphologically similar [26]. However, it is not excluded that the invader could facilitate the occurrence of some dissimilar native species [26].

Various competitive interactions can affect species biomass and contribute to invasion processes [27]. In stressful situations, such as competition, plants may exhibit differential biomass allocation, prioritizing either aerial (or aquatic, in the case of aquatic organisms) or underground parts [28]. For macrophytes, differential resource allocation is related to the pursuit of greater plant development, such as nutrient assimilation and leaf growth, in response to stressful environmental variations [29,30,31].

*H. verticillata* is an invasive submerged aquatic macrophyte [32] that grows in a variety of environments, from lentic to lotic, with great adaptability [3,33]. It has higher energy density compared to native species occupying the same niche, becoming a resource for herbivores [34]. Native to Asia and Australia, it is found on every continent, causing problems for aquatic ecosystems worldwide [35]. In Brazil, it exhibits significant dominance in several river basins [36]. It is often confused with the native species *Egeria najas* Planch. and *Egeria densa* Planch. due to phylogenetic and functional similarities, as all three belong to the Hydrocharitaceae family [3]. *E. najas* and *E. densa* are submerged macrophyte species native to Brazil. Frequently confused with the invasive *H. verticillata*, these natives also cause significant impacts due to excessive growth in reservoirs [37]. They have simple adventitious roots [38] and respond negatively to reservoir disturbances during dry periods [39]. *E. densa* has a higher light requirement and is more affected by water turbidity, while *E. najas* shows greater tolerance and thrives better in reservoirs [40,41]. *H. verticillata* possesses great phenotypic plasticity and disperses efficiently via propagules [42,43,44]. The large biomass exhibited by this species in dominant situations can displace native species and alter their occurrence in invaded areas [26,36].

Understanding the interactions between native and invasive exotic species is imperative for controlling their spread and developing more effective mitigation strategies [45]. Exploring functional traits related to biomass and differential resource allocation capabilities can reveal important competitive abilities for plant dominance and are crucial for the persistence of exotic species after invasion [46]. Studies on submerged macrophytes conducted in tropical mesocosms, which allow for the control and isolation of desirable or undesirable variables in near-natural conditions, are essential to advancing knowledge about the biology and ecology of these plants [47,48]. This approach facilitates the study of complex species interactions in a controlled environment that simulates natural conditions [49,50].

In this study, we aim to evaluate the effects of native submerged aquatic macrophyte (SAM) species on the invasibility of *H. verticillata*. In a mesocosm experiment (i.e., exploring local climatic conditions), we subjected the invader to growth under (A) intraspecific competition (Control) and (B) interspecific interaction with two native SAM species. Our hypothesis is that competition will negatively influence the development of invasive exotic species. Thus, *H. verticillata* would exhibit changes in leaf and subterranean biomass patterns when subjected to competition with different native SAM species, compared to its development under intraspecific competition.

## 2. Results

Throughout the experiment period, the daily average temperature was 17.71 °C, with a minimum average of 13.45 °C and a maximum average of 20.42 °C, average humidity of 79.40%, and average precipitation of 5.05 mm, according to data from the meteorological station of Simepar [51], located near the Piracema Park Station, Itaipu Hydroelectric (25°25′46″ S, 54°34′50″ W). The initial biomass of 10 *H. verticillata* ramets was 0.58 g (SD = 0.09), and therefore, the initial biomass of each ramet was 0.06 g.

### 2.1. Total Biomass of Submerged Aquatic Macrophytes (SAMs)

In terms of total biomass (sum of aquatic and subterranean biomass) (Figure 1), the intraspecific treatment had an average biomass of 122.93 g (SD = 30.65), and the interspecific treatment had an average biomass of 113.40 g (SD = 13.36). These results suggest that, although interspecific competition with native species slightly reduces the total biomass of *H. verticillata*, the effect is not strong enough to result in significant differences. This indicates that the invasive species maintains its competitive potential even in the presence of native species. Further analysis of foliar and subterranean biomass components provides additional insights into resource allocation patterns (Table 1).

### 2.2. Subterranean Biomass of Submerged Aquatic Macrophytes

The subterranean biomass of SAMs (Figure 2) showed that the intraspecific treatment had an average biomass of 70.58 g (SD = 19.22), compared to the interspecific treatment, which had an average biomass of 37.35 g (SD = 11.39; GLM; z = −3.073; *p* = 0.006) (Figure 2, Table 1). These results indicate that interspecific competition with native species reduced the subterranean biomass of *H. verticillata* compared to intraspecific competition, highlighting the impact of the presence of native species on the allocation of subterranean biomass. The significant difference between the treatments suggests that competition with native species has a greater effect on subterranean biomass than on the total or foliar biomass of *H. verticillata*.

### 2.3. Total Foliar Biomass of Macrophytes

As for the total foliar biomass of SAMs (Figure 3), the interspecific treatment had an average biomass of 76.05 g (SD = 13.75), presenting a higher mean value than the intraspecific treatment, which had an average biomass of 52.35 g (SD = 10.45; GLM; Z = 3.293; *p* = 0.002). These results suggest that interspecific competition with native species promoted the allocation of more biomass to foliar structures in *H. verticillata* compared to intraspecific competition. The statistical analysis confirmed a significant difference between the two treatments, highlighting the role of interspecific competition in shaping the biomass distribution towards foliar structures.

### 2.4. Foliar Biomass Increment of H. verticillata

In terms of *H. verticillata* biomass increment (Figure 4), the interspecific treatment (3103%; SD = 519) had a higher mean value than the intraspecific treatment (1003%; SD = 200; GLM; Z = 9.782; *p* = 0.000). These data indicate that, under interspecific competition, *H. verticillata* had a substantial increase in its foliar biomass, suggesting a competitive adaptation response to maximize leaf growth at the expense of root growth.

## 3. Discussion

The study tested the effects of native submerged macrophytes’ competitiveness on the growth of the invasive exotic species *H. verticillata*. Our results demonstrate that *H. verticillata* altered its development by promoting differential biomass allocation in response to changes in intraspecific and interspecific competition in the mesocosm treatments. When subjected to interspecific competition, this invasive species allocated biomass to foliar structures at the expense of root structures. It was observed that the native submerged species did not act as a barrier but influenced a reduction in root biomass and an increase in foliar biomass. This effect may be attributed to interspecific competitive interactions, which influence the performance of invasive plants [52,53,54,55,56].

Communities with higher diversity tend to be more competitive and resistant to invasion, which results in greater success in suppressing invasive species [14,54,55]. Furthermore, it is highlighted that communities with high diversity may contain a significant proportion of exotic species, with this diversity being maintained by environmental heterogeneity and interactions between native and exotic species [56]. In other cases, even with diversity, these areas may suffer from invasive species suppression, which can affect small spaces without exerting pressure on large-scale diversity [57], affecting species richness and functional diversity, though with a certain resilience of seeds when confronted by invasive species [58].

Spatial scales must be considered in invasion models, as abiotic factors influence the success of the invasive macrophyte *H. verticillata* at different spatial scales. At a smaller spatial scale, organic matter becomes important, and competitive interactions may inhibit invasion [11]. Therefore, biodiversity loss can increase the vulnerability of native communities to invasions [14], affecting the structure and functioning of ecosystems.

Resource allocation is the foundation for adopting different behavioral strategies in response to environmental pressures [59]. Morphological variations and decisions related to biomass allocation are crucial for plants’ developmental success, allowing them to efficiently obtain resources during competition [60,61] and extend their residence time [62]. In this context, we believe that the invasive species prioritized vegetative growth in photosynthetic and dispersal structures upon detecting competition with native plants. Limiting factors, such as light and space, may intensify competition between species, leading to the adoption of strategies for resource acquisition, thereby promoting a positive differential allocation towards the growth of photosynthetic structures [63].

However, it is important to emphasize that this differential resource allocation can favor certain structures at the expense of others [64]. It is not yet certain whether this strategy will have positive or negative effects on the invasive species in the long term, as available results come from six-month experiments conducted with perennial plants [32]. This highlights the need for longer-duration experiments that consider environmental variations with different complexities. It is essential to understand that ecosystem changes may be the main factor influencing interactions related to differential allocations, impacting communities by modifying the availability of limited resources and the allocation priorities of each species [65].

Differential allocation has been evidenced in invasive exotic species, which demonstrate biomass allocation towards foliar areas to enhance light acquisition and photosynthetic capacity [66]. During the invasion process, this differential allocation primarily occurs when species are in interspecific competition, increasing their growth regardless of correlated environmental issues [67]. In some cases, the invasive species adopts this strategy in response to the synergistic effect of interspecific competition with native species and environmental gradients [68]. In other cases, interspecific competition may result in reproductive allocations due to changes in abiotic conditions caused by interactions with native species [69]. This functional characteristic is one of the determinants of the potential for macrophyte species to become invasive [70,71].

Our results confirm the differential allocation of biomass by the invasive species under interspecific competition. With increased competitive potential in seeking vegetative growth to surpass native species, the invasive species compromised its root structures, resulting in reduced resource allocation for anchoring and storage. The increase in foliar biomass of *H. verticillata* at the expense of its subterranean structures indicates that the invasive species exhibited a differentiated growth strategy in the presence of natives. The presence of natives differentially influenced the development of the invasive species, which indeed recognized a competitor. Despite the limitation of its root structure, the greater foliar biomass under interspecific competition may confer competitive advantages to the invasive species in terms of light acquisition and vegetative propagation. During the experiment period, under ideal light and substrate nutrient conditions, the exotic species maintained significant biomass growth above the sediment despite the reduction in root growth, suggesting a trade-off. Continuing this growth pattern could eventually negatively impact the available aquatic space for native species. Thus, the invasive species could gain advantages in vegetative dissemination. However, this potential advantage is inconclusive. The reduction in *H. verticillata* roots may have consequences for its ability to anchor to the substrate and obtain nutrients, especially in more challenging environments [72,73,74]. As such, the balance of advantages and disadvantages promoted by differential allocation should be explored in future studies.

The 50% shading used in the mesocosm experiments to avoid algae proliferation and water overheating also deserves attention. This experimental choice, based on the literature, may have influenced the development of *H. verticillata* and native species, suggesting the need to consider its possible effects on the competitive dynamics between species. Thus, we suggest that future studies consider different shading levels in small-scale experiments in tropical regions and their potential interactions with root growth dynamics, as well as evaluate the impact of other abiotic factors (i.e., nutrient availability and temperature variations).

Despite these considerations, our contributions are relevant for understanding the competitive interactions between *H. verticillata* and native species, both above and below the substrate. By identifying that *H. verticillata* increases foliar biomass but reduces root growth, management practices focusing on root structures are suggested. The removal of ramets could offer promising results, as the potential for regeneration through root structures may be compromised due to the significantly reduced biomass. The study reaffirms the importance of environmental management strategies that consider these multiple factors, as well as the need for further research exploring the complex interactions between invasive and native species under different ecological conditions, the effects of root structure reduction on invasiveness and competition in submerged aquatic plants, considering temporal variations, and incorporating greater complexity into experimental communities and other native community parameters, such as species composition, abundance, and functional traits, which should be explored as potential barriers to invasion under different environmental factor variations.

## 4. Materials and Methods

### 4.1. Collection Area of SAMs

The Itaipu Reservoir, located on the Paraná River between Brazil and Paraguay, has relatively stable hydrometric levels on its eastern side (Brazilian side), with an average depth of 22 m and water quality ranging from oligotrophic to eutrophic [75]. Its large expanse of shallow areas (between 0.5 and 4.0 m deep) favors the development of submerged macrophytes [11]. The macrophytes were collected from two of the eight tributary rivers formed by the Itaipu reservoir that had already been monitored [76], the Passo Cuê River (25°20′19″ S, 54°26′41″ W) and the São João River. (25°04′21″ S, 54°22′27″ W) (Figure 5). Using a rake, we collected samples of the native species *E. najas*, *E. densa*, and the exotic invasive *H. verticillata*, Gathering the collection from both arms for the mesocosm experiment. These species are among the main submerged macrophytes found in the reservoir, where four species of Hydrocharitaceae can be found, contributing to a total diversity of 87 macrophyte species, five of which are non-native [37].

### 4.2. Plant Processing and Mesocosm Setup

The collected macrophytes were stored in plastic bags with reservoir water. In the laboratory, the macrophytes were carefully washed to remove macroinvertebrates, algae, and debris. The experiment was conducted over six months at the Parque da Piracema station, located at the Itaipu Hydroelectric (25°25′46″ S, 54°34′50″ W), in 10 cylindrical tanks of 500 L capacity, 60 cm in height, and 1 m in diameter, installed outdoors with 50% shading, ideal for cultivating submerged macrophytes [69]. Each tank contained a 25 kg layer of industrialized organic substrate and a 2 cm layer of coarse sand and was filled with well water (without the addition of artificial chemical components). The macrophyte ramets were standardized into 15 cm fragments for planting, all on the same day. During the experiment, partial water exchanges (50%) were performed weekly to prevent algae growth (Figure 6). To ensure that the observed effects were solely due to plant competition, the experiment was conducted using the same substrate, space, and number of initial ramets across the different treatments, controlling the main environmental variables responsible for plant development and providing standardized conditions across treatments. However, the initial number of *H. verticillata* ramets varied for each treatment. To address this analytically, we worked with the biomass acquired throughout the experiment, allowing us to make balanced comparisons of invasive development, regardless of the treatment and initial number of ramets, while maintaining standardized treatments.

To assess the influence of different species quantities on the invader’s development, the experiment was conducted with treatments involving different interactions, randomly distributed across the 10 tanks (5 replicates per treatment) as follows: (1) Control treatment of intraspecific interaction, planting 90 ramets of *H. verticillata*, to evaluate the invader’s development without the influence of interspecific competition; and (2) Interspecific interaction treatment, planting 90 ramets, with 30 ramets of *H. verticillata*, 30 ramets of *E. najas*, and 30 ramets of *E. densa*.

To estimate the initial biomass of *H. verticillata* ramets, 50 standardized ramets (15 cm) were separated and dehydrated in an oven at 45 °C until constant weight was achieved (about 2 days). The dry mass (g) was obtained gravimetrically by separating the ramets into five sets of 10. Thus, the estimated initial biomass of each *H. verticillata* ramet is the average dry mass obtained (average of five sets of 10 dried ramets) divided by 10. The initial biomass value of *H. verticillata* in each treatment is the estimated initial weight of each ramet multiplied by the number of ramets of the invader at the start of the experiment.

### 4.3. Data Collection

The experiments were dismantled after six months of growth of the macrophyte plants. All plant material was removed from the mesocosms, and the ramets (aquatic biomass) were separated from the roots (subterranean biomass) in each replicate for each treatment. The ramets were also sorted by species. It was not possible to separate the roots by species, as by the end of the experiment, all root structures were entangled. In the laboratory, the biomass (ramets and roots) was thoroughly washed to remove algae and invertebrates and then dried in an oven at 45 °C until a constant weight was achieved (about 4 days). The dry mass (g) of each set (total roots, ramets by species in each replicate of each treatment) was obtained gravimetrically.

### 4.4. Data Analysis

To assess the biomass increase in *H. verticillata* ramets in each treatment over the course of the experiment, we divided the dry biomass of the invader obtained gravimetrically at the end of the experiment by the initial biomass and calculated the percentage increase. To assess the effects of competitive relationships of *H. verticillata* on preferential biomass allocation (above the sediment or subterranean biomass) in the competitive treatments, we structured the data analysis by comparing (1) differences in total biomass per treatment (biomass of ramets and root structures combined for all species per treatment); (2) differences in subterranean biomass per treatment (biomass of all root structures combined); (3) differences in foliar biomass per treatment (biomass above the sediment, combined for all species); and (4) differences in the increase in foliar biomass of the invasive species per treatment (increase in the biomass of *H. verticillata* ramets above the sediment).

For this, a generalized linear model was used with the glm function from the R software 2024.04.1+748 [77]. To compare each treatment pairwise, we used a multiple comparisons test for general linear hypotheses by Tukey with the glht function from the multcomp package [78]. The graphs were generated using the Excel software [79].

## 5. Conclusions

The results indicated that the presence of native species influenced the biomass allocation pattern of *Hydrilla verticillata*, prioritizing the growth of foliar structures at the expense of root structures, particularly under interspecific competition. Although the native species did not function as a complete barrier to the dominance of the invasive species, they induced a reduction in the root biomass of *H. verticillata*, which may compromise its anchorage and ability to exploit nutrients in the substrate. These findings suggest that, while the hypothesis was partially supported, further studies are needed to fully understand the long-term implications of reduced root structures and their impact on aquatic plant invasion. Therefore, the effect of native species diversity on invasion dynamics remains complex and dependent on factors such as competition intensity and environmental conditions.

## Figures and Tables

**Figure 1 plants-13-03500-f001:**
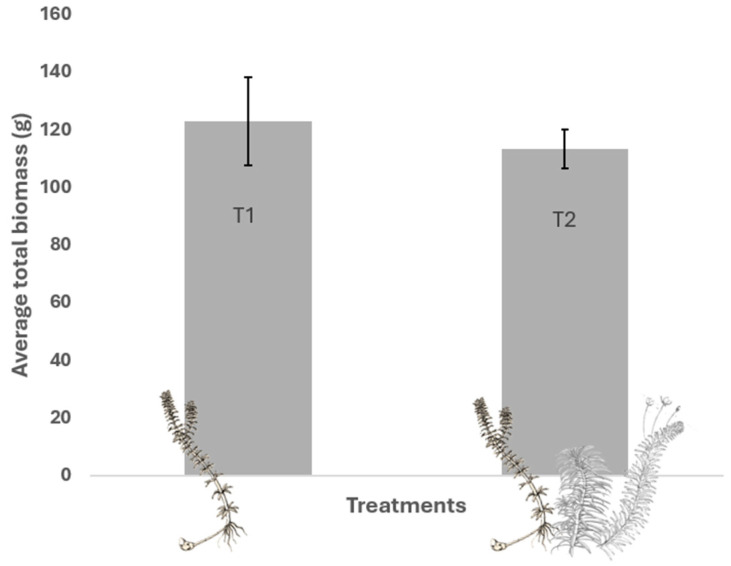
Mean total biomass (foliar + subterranean) of SAMs at the end of the experiment under different competitive treatments. T1 = control with intraspecific competition of *H. verticillata* only; T2 = with interspecific competition of *H. verticillata*, *E. najas*, and *E. densa*.

**Figure 2 plants-13-03500-f002:**
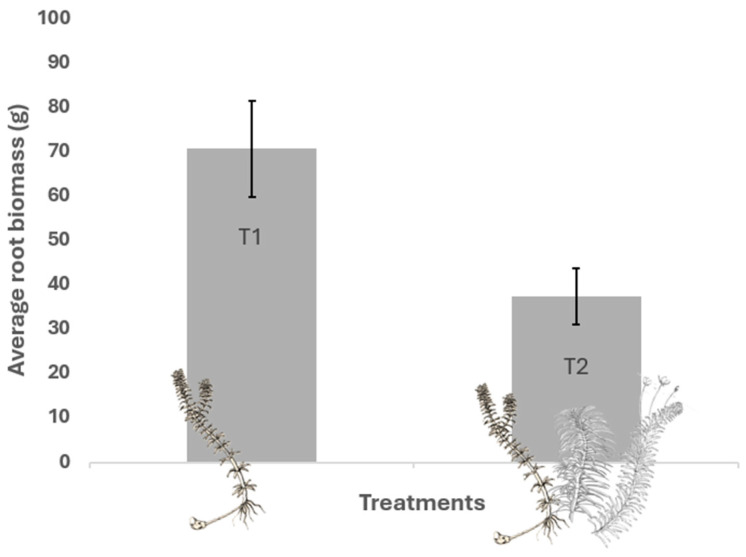
Mean subterranean biomass of SAMs at the end of the experiment under different competitive treatments. T1 = control with intraspecific competition of *H. verticillata* only; T2 = with interspecific competition of *H. verticillata*, *E. najas*, and *E. densa*.

**Figure 3 plants-13-03500-f003:**
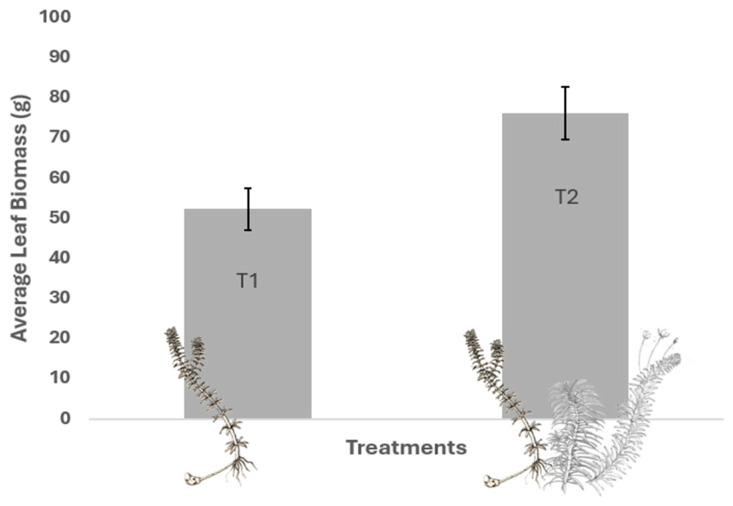
Mean foliar biomass of SAMs at the end of the experiment under different competitive treatments. T1 = control with intraspecific competition of *H. verticillata* only; T2 = with interspecific competition of *H. verticillata*, *E. najas*, and *E. densa*.

**Figure 4 plants-13-03500-f004:**
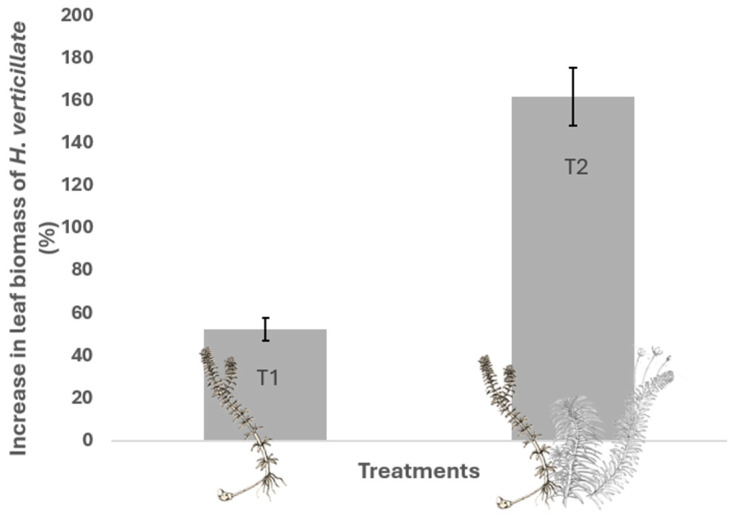
Foliar biomass increment (in percentage) of *H. verticillata* at the end of the experiment under different competitive treatments. T1 = control with intraspecific competition of *H. verticillata* only; T2 = with interspecific competition of *H. verticillata*, *E. najas*, and *E. densa*.

**Figure 5 plants-13-03500-f005:**
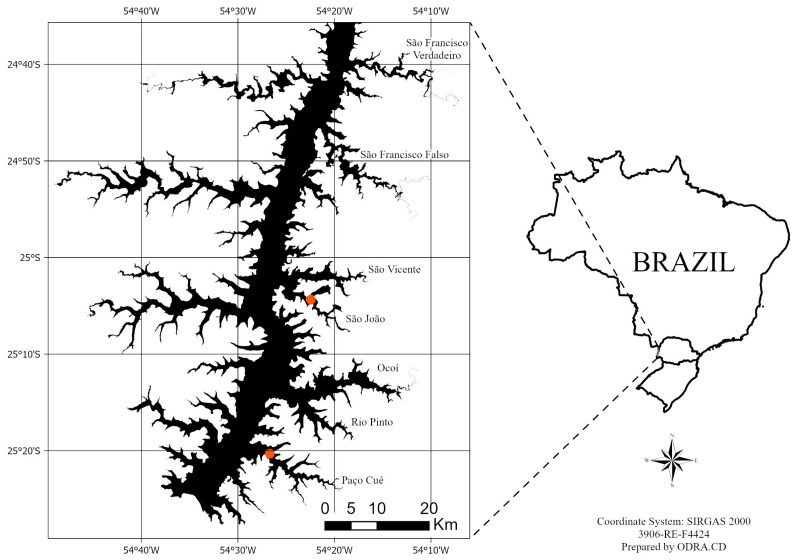
Collection area in the Itaipu Reservoir, located in the Paraná River basin. The marked points are the Passo Cuê and São João arms, where the invasive exotic *H. verticillata* and the native species *E. najas* and *E. densa* were collected.

**Figure 6 plants-13-03500-f006:**
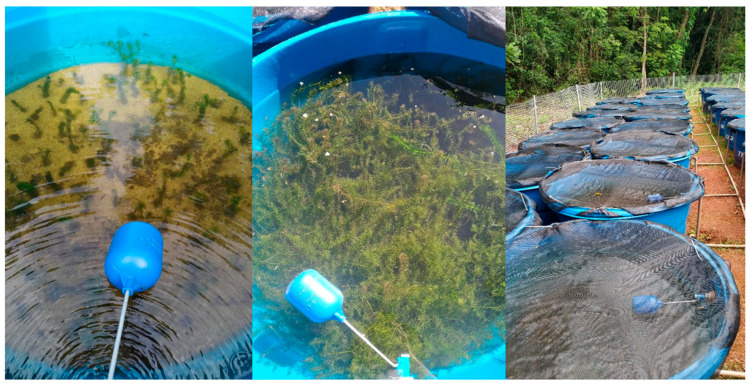
Mesocosms in 500 L tanks; newly planted macrophytes; macrophytes after five months; 50% shading setup.

**Table 1 plants-13-03500-t001:** Post hoc Tukey pairwise test results for the generalized linear model (GLM) comparing the biomass growth of SAMs (total, subterranean, aquatic, and biomass increment) in different competitive treatments, focusing on the invasive species *H. verticillata*. T1 = control with intraspecific competition of *H. verticillata*; T2 = with interspecific competition of *H. verticillata*, *E. najas*, and *E. densa*.

Treatment	Total Biomass		Subterranean Biomass		Foliar Biomass		*H. verticillata* Biomass Increment	
	(Tukey Model)		(Tukey Model)		(Tukey Model)		(Tukey Model)	
	Estimate	z-Value	Estimate	z-Value	Estimate	z-Value	Estimate	z-Value
T1 × T2	−0.08	−0.63	−0.63	−3.07 **	0.37	3.29 **	2100.2	9.782 ***

** < 0.01; *** < 0.001.

## Data Availability

All the data discussed in the article.

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
