# Peer review of "Growth Allocation Shifts in the Invasive *Hydrilla verticillata* Under Interspecific Competition with Native Submerged Macrophytes"

_plants, 2024, doi:10.3390/plants13243500_

Round 1
Reviewer 1 Report
Comments and Suggestions for Authors
The manuscript presents a very interesting experimental study of the interaction between the invasive species Hydrilla verticillata and native macrophytes in the Itaipu Reservoir, located on the Paraná River between Brazil and Paraguay using mesocosms. Using a six-month experiment, the authors revealed how macrophyte growth strategies changed.This knowledge is important both for understanding the biology of species and for developing ways to control the condition of water bodies.
The biggest question for the authors arises when reading the Results section of the manuscript. Firstly, the results are generally very poorly described. For example, the biomass of Hydrilla verticillata species itself is not presented for interspecific competition. There are only data on the total biomass of all macroalgae. How can the authors conclude on the basis of these data: Under conditions of interspecific competition, the invasive species allocated more biomass to aquatic structures at the expense of fixation, absorption, and storage structures.
In addition, the perception of the manuscript can be improved by adding illustrative material, such as a photo of the mesocosm.
A description of the application of the gravimetric method in the work would be nice to see.
In Figures 1-4, it is necessary to add the labels T1, T2 or to add the labels a and b to the caption.
Author Response
Dear Reviewer 1,
We sincerely thank you for your detailed and thoughtful feedback on our manuscript. Your insightful comments and constructive suggestions have significantly contributed to improving the quality and clarity of our work. Your feedback has been invaluable in refining our manuscript, and we greatly appreciate the time and effort you dedicated to reviewing our work.
We are submitting a document with all the changes marked in track mode to facilitate the review process and highlight the revised sections, along with the updated manuscript.Thank you once again for your contributions to this study.
The manuscript presents a very interesting experimental study of the interaction between the invasive species Hydrilla verticillata and native macrophytes in the Itaipu Reservoir, located on the Paraná River between Brazil and Paraguay using mesocosms. Using a six-month experiment, the authors revealed how macrophyte growth strategies changed.This knowledge is important both for understanding the biology of species and for developing ways to control the condition of water bodies.
Comments 1: The biggest question for the authors arises when reading the Results section of the manuscript. Firstly, the results are generally very poorly described. For example, the biomass of Hydrilla verticillata species itself is not presented for interspecific competition. There are only data on the total biomass of all macroalgae. How can the authors conclude on the basis of these data: Under conditions of interspecific competition, the invasive species allocated more biomass to aquatic structures at the expense of fixation, absorption, and storage structures.
Response 1: L 117, 120-134. We appreciate your constructive comments. In response to your observation regarding the Results section, we have made the necessary changes to improve the description of the results. We believe these revisions provide a clearer understanding of the findings.
Comments 2: In addition, the perception of the manuscript can be improved by adding illustrative material, such as a photo of the mesocosm.
Response 2: L 345. We appreciate the suggestion to include illustrative material to improve the clarity of the manuscript. In response, we have added a figure with three images of the mesocosm to enhance the presentation of the experimental setup. We believe this addition will help readers better visualize the conditions under which the experiment was conducted.
Comments 3: A description of the application of the gravimetric method in the work would be nice to see.
response 3: Thank you for your suggestion.
Comments 4: In Figures 1-4, it is necessary to add the labels T1, T2 or to add the labels a and b to the caption.
Response 4: Thank you for the suggestion. We have included "T1" and "T2" in the graphs to make it clearer which treatments they refer to.
Sincerely,
Letícia da Costa
On behalf of all co-authors

Reviewer 2 Report
Comments and Suggestions for Authors
Comments
This article evaluates the study “Growth Allocation Shifts in the Invasive Hydrilla verticillata (L.f.) Royle Under Interspecific Competition with Native Sub-merged Macrophytes”. This manuscript is interesting and information could be helpful in providing the Invasiveness of Hydrilla verticillata.
I have some comments and suggestion related to manuscript.
Title:
The title is looking good but mentioning full scientific name along with authority is usually not preferred so my suggestion for title is “Growth Allocation Shifts in the Invasive Hydrilla verticillata Under Interspecific Competition with Native Sub-merged Macrophytes”
Abstract
Abstract lack methodology and results as the authors provide more information about the background and future concerns without providing the outcomes of the study.
Try to add some background information and then add problem statement, objective, methodology, results and conclusion.
The abstract also lacks methodology adopted to conduct study.
The last sentence of this section needs to improve by adding future implications.
Introduction
This section is too lengthy. Try to shorten the first four paragraphs (L 33-67).
Mention English name as well at first use for each species.
Use E. densa instead of using full name after first time of using it as full and follow it in the whole manuscript.
Use same for other species as well.
Results
This section needs to extend by mentioning the proper impact of inter and intraspecific competition. Mentioning the highest and lowest observed value for each parameter.
Also, mention the source and figure of the climate data presented L 118-120.
Table 1 mentioning values by using comma (0,37) and this is not acceptable as it is not according to SI rules. These values must be like this (0.37).
Figure 1-4: Make axes bold and increase visibility by properly mentioning treatments and values in both axes.
Discussion
This section is looking good and providing sufficient support to the results
Material and method
Provide quadrants for the selection location is necessary.
Author provide information about the sampling done from two arms. But I could not seed further explanation. The data from both arms were pooled or what?
Use a passive approach as author mention we performed------- again and again that is not good.
Moreover how the data was analyzed? Which study design and which software was used for data analysis?
Conclusion
Provide the exact outcome of the study fulfilling the objectives and then briefly explain the outcomes.
Use space between value and unit and follow it in the whole manuscript.
Italicize the scientific name used in the whole manuscript.
At first use, write English name and scientific name with authority and then use English or Scientific name in the whole manuscript by following same pattern.
Use proper symbol for mentioning units
Author Response
Dear Reviewer 2,
We deeply appreciate your detailed review and valuable suggestions for improving our manuscript, "Growth Allocation Shifts in the Invasive Hydrilla verticillata Under Interspecific Competition with Native Submerged Macrophytes." Your comments have been instrumental in refining the quality and presentation of our work. We hope that these revisions meet the reviewer's expectations. We remain available for any further clarifications or additional adjustments that may be necessary.
We are submitting a document with all the changes marked in track mode to facilitate the review process and highlight the revised sections, along with the updated manuscript. Thank you again for your time and effort in reviewing our work. Your thoughtful comments have greatly enhanced the manuscript, and we sincerely appreciate your contributions.
Hydrilla verticillata (L.f.) Royle Under Interspecific Competition with Native Sub-merged Macrophytes”. This manuscript is interesting and information could be helpful in providing the Invasiveness of Hydrilla verticillata.
I have some comments and suggestion related to manuscript.
Comments 1: Title:
- The title is looking good but mentioning full scientific name along with authority is usually not preferred so my suggestion for title is “Growth Allocation Shifts in the Invasive Hydrilla verticillata Under Interspecific Competition with Native Sub-merged Macrophytes”
Response 1: L 03. Thank you for your suggestion. We have revised the title as per your recommendation. The new title is:
“Growth Allocation Shifts in the Invasive Hydrilla verticillata Under Interspecific Competition with Native Submerged Macrophytes.”
Comments 2: Abstract
- Abstract lack methodology and results as the authors provide more information about the background and future concerns without providing the outcomes of the study.
- Try to add some background information and then add problem statement, objective, methodology, results and conclusion.
- The abstract also lacks methodology adopted to conduct study.
- The last sentence of this section needs to improve by adding future implications.
Response 2: L 14-33. Thank you for the suggestion. We have added more detailed information about the study's methodology and results. The abstract now includes a description of the treatments, experimental conditions, and key quantitative results, such as the differences in foliar and subterranean biomass between treatments and the biomass increment of H. verticillata. We also revised the conclusion to include future implications related to the reduction of subterranean structures and the persistence of the invasive species.
The revised version of the abstract follows the suggested structure:
- Context and problem;
- Study objective;
- Methodology and experiment description;
- Key results with quantitative data;
- Conclusion and future implications.
Comments 3: Introduction
- This section is too lengthy. Try to shorten the first four paragraphs (L 33-67).
- Mention English name as well at first use for each species.
- Use E. densa instead of using full name after first time of using it as full and follow it in the whole manuscript.
- Use same for other species as well.
Response 3: L 38-55, L 109. Thank you for your valuable feedback. The opening paragraphs have been shortened and revised while retaining the main information and the common English names for each species have been included upon their first mention in the text.
Comments 4: Results
- This section needs to extend by mentioning the proper impact of inter and intraspecific competition. Mentioning the highest and lowest observed value for each parameter.
- Also, mention the source and figure of the climate data presented L 118-120.
- Table 1 mentioning values by using comma (0,37) and this is not acceptable as it is not according to SI rules. These values must be like this (0.37).
- Figure 1-4: Make axes bold and increase visibility by properly mentioning treatments and values in both axes.
Response 4: L 119, L 128-134, L 155-162, L 175-181, L 187-189. Thank you for your valuable feedback. In the results section, we expanded on the impact of interspecific competition, as requested, by including the highest and lowest values observed for each parameter. Specifically, we briefly discussed the effects of competition on the biomass of submerged aquatic macrophytes to better express them in the discussion section. We highlighted the differential biomass allocation in growth strategies, particularly with evident costs in subterranean biomass allocation, indicating that H. verticillata responded differently to competitive conditions by allocating biomass in varied ways depending on the treatment.
Comments 5: Discussion
- This section is looking good and providing sufficient support to the results
Response 5: Thank you for the positive feedback regarding the Discussion section. We are glad that the content is clear and provides sufficient support to the results.
Comments 6: Material and method
- Provide quadrants for the selection location is necessary.
- Author provide information about the sampling done from two arms. But I could not seed further explanation. The data from both arms were pooled or what?
- Use a passive approach as author mention we performed------- again and again that is not good.
- Moreover how the data was analyzed? Which study design and which software was used for data analysis?
Response 6: We would like to thank the reviewer for their valuable comments and suggestions.
L 295-300, L 311-327, L 330-338. In relation to the suggestion of providing quadrants for the location selection, we clarify that the plants collected at the two points were grouped to be transplanted into the mesocosms. The selection of these points in the rivers was made randomly, as they had already been previously monitored in a prior study. We chose these locations due to the higher likelihood of finding the macrophytes, ensuring greater precision and reliability in the collection.
L 374-378. We revised the manuscript to adopt a more passive approach, as recommended, in order to avoid the repetitive use of the phrase "we performed," and provided more details about the data analysis. Specifically, we described the study design and clarified that the data were analyzed using a generalized linear model (GLM) in R software (R Core Team, 2024).
We appreciate your suggestions and believe that the manuscript is now clearer and more complete as a result.
Comments 7: Conclusion
- Provide the exact outcome of the study fulfilling the objectives and then briefly explain the outcomes.
- Use space between value and unit and follow it in the whole manuscript.
- Italicize the scientific name used in the whole manuscript.
- At first use, write English name and scientific name with authority and then use English or Scientific name in the whole manuscript by following same pattern.
- Use proper symbol for mentioning units
Response 7: L 381-391. Thank you for your valuable feedback. In the conclusion section, we provide the results of our study that meet the objectives. We then briefly explain the findings, focusing on how the data support our hypothesis and contribute to the understanding of the ecological dynamics of Hydrilla verticillata in response to interspecific competition.
Sincerely,
Letícia da Costa
On behalf of all co-authors

Reviewer 3 Report
Comments and Suggestions for Authors
The manuscript "Growth allocation in the invasive Hydrilla verticillata (L.f.) Royle under interspecifix competition with native submerged macrophytes" is interesting, but unclear, with many mistakes and needs substantial revisions.
L: 37 - "interact with climate change" - What does it mean? Ot is scientific jargon. Please, change it to "effect on climate change" and describe the way.
L: 52-53 - the sentence: "Experiments suggest that low phylogenetic diversity among native species facilitates the occurrence of exotics, and that exotic species 53 closely related to natives are potentially more invasive" should be changes to "Experiments suggest that low native species diversity facilitates the occurrence of exotics....."
L: 113 - change "our prediction" to "our hypothesis"
Figure 1 - The Authors state: "Figure 1. Mean global biomass (foliar + subterranean) of SAMs at the end of the experiment under different competitive treatments. T1 = control with intraspecific competition of H. verticillata only; T2 = with interspecific competition of H. verticillata, E. najas, and E. densa." My questions: Where are "T1" and "T2" in the figure? These terms are only in the description. "Global biomass" or "total biomass" - please, clarify. What does mean "a" and "b" in the figure?
Figure 2 - The Author state: "Figure 2. Mean subterranean biomass of SAMs at the end of the experiment under different competitive treatments. T1 = control with intraspecific competition of H. verticillata only; T2 = with interspecific competition of H. verticillata, E. najas, and E. densa. Different letters represent significant differences in mean biomass according to GLM."
My questions: Where are "T1" and "T2" in the figure? These terms are only in the descriptions.
Figure 3 - The Author state: "Figure 3. Mean foliar biomass of SAMs at the end of the experiment under different competitive 166 treatments. T1 = control with intraspecific competition of H. verticillata only; T2 = with interspecific 167 competition of H. verticillata, E. najas, and E. densa. Different letters represent significant differences 168 in mean biomass according to GLM." but in the figure we can see "average leaf biomass". Where are "T1", "T2".
Figure 4 - The Authors state: "Figure 4. Foliar biomass increment (in percentage) of H. verticillata at the end of the experiment 178 under different competitive treatments. T1 = control with intraspecific competition of H. verticillata 179 only; T2 = with interspecific competition of H. verticillata, E. najas, and E. densa." but we can see "increase in leaf biomass". What does mean "a"? What does mean "b"?
L 185: The scientific name Hydrilla verticillata should be written full only the first time in the text. Please, correct it, and in the case of other species.
Please, change the term "global biomass" to "total biomass" in the whole manuscript.
Author Response
Dear Reviewer 3,
We hope that these revisions meet the reviewer's expectations. We remain available for any further clarifications or additional adjustments that may be necessary. Your insightful observations and detailed recommendations have greatly improved the manuscript. We are deeply grateful for your time, effort, and expertise in reviewing our work.
We are submitting a document with all the changes marked in track mode to facilitate the review process and highlight the revised sections, along with the updated manuscript. Thank you again for your time and effort in reviewing our work.
The manuscript "Growth allocation in the invasive Hydrilla verticillata (L.f.) Royle under interspecifix competition with native submerged macrophytes" is interesting, but unclear, with many mistakes and needs substantial revisions.
Comments 1: L: 37 - "interact with climate change" - What does it mean? Ot is scientific jargon. Please, change it to "effect on climate change" and describe the way.
Response 1: Thank you for the suggestion. We have made the requested change, replacing the term "interact with climate change" with "effect on climate change," which more accurately reflects the context and makes the explanation clearer for readers.
Comments 2: L: 52-53 - the sentence: "Experiments suggest that low phylogenetic diversity among native species facilitates the occurrence of exotics, and that exotic species 53 closely related to natives are potentially more invasive" should be changes to "Experiments suggest that low native species diversity facilitates the occurrence of exotics....."
Response 2: L 50. Thank you for the suggestion. We agree that simplifying the sentence makes the message clearer and more direct. Therefore, we have made the requested change and rewritten the sentence to reflect the diversity of native species, as indicated.
Comments 3: L: 113 - change "our prediction" to "our hypothesis"
Response 3: L 109. Thank you for the suggestion. We agree that the term "hypothesis" is more appropriate for the scientific context and more accurately reflects the study's intention. Therefore, we have made the requested change.
Comments 4: Figure 1 - The Authors state: "Figure 1. Mean global biomass (foliar + subterranean) of SAMs at the end of the experiment under different competitive treatments. T1 = control with intraspecific competition of H. verticillata only; T2 = with interspecific competition of H. verticillata, E. najas, and E. densa." My questions: Where are "T1" and "T2" in the figure? These terms are only in the description. "Global biomass" or "total biomass" - please, clarify. What does mean "a" and "b" in the figure?
Response 4: Thank you for the suggestion. We have included "T1" and "T2" in the graphs to make it clearer which treatments they refer to.
We agree that the term "global biomass" could cause confusion. We have changed the expression to "total biomass," which is more precise and commonly used in the scientific literature.
We have decided to exclude the symbols "a" and "b" from the figure, as we considered that this distinction was not essential for the understanding of the results. The terms "a" and "b" referred to the different statistical comparison groups used to indicate where significant differences occurred between the treatments.
Comments 5: Figure 2 - The Author state: "Figure 2. Mean subterranean biomass of SAMs at the end of the experiment under different competitive treatments. T1 = control with intraspecific competition of H. verticillata only; T2 = with interspecific competition of H. verticillata, E. najas, and E. densa. Different letters represent significant differences in mean biomass according to GLM."
My questions: Where are "T1" and "T2" in the figure? These terms are only in the descriptions.
Response 5: Thank you for the suggestion. We have included "T1" and "T2" in the graphs to make it clearer which treatments they refer to.
Comments 6: Figure 3 - The Author state: "Figure 3. Mean foliar biomass of SAMs at the end of the experiment under different competitive 166 treatments. T1 = control with intraspecific competition of H. verticillata only; T2 = with interspecific 167 competition of H. verticillata, E. najas, and E. densa. Different letters represent significant differences 168 in mean biomass according to GLM." but in the figure we can see "average leaf biomass". Where are "T1", "T2".
Response 6: Thank you for the suggestion. We have included "T1" and "T2" in the graphs to make it clearer which treatments they refer to.
Comments 7: Figure 4 - The Authors state: "Figure 4. Foliar biomass increment (in percentage) of H. verticillata at the end of the experiment 178 under different competitive treatments. T1 = control with intraspecific competition of H. verticillata 179 only; T2 = with interspecific competition of H. verticillata, E. najas, and E. densa." but we can see "increase in leaf biomass". What does mean "a"? What does mean "b"?
Response 7: We have decided to exclude the symbols "a" and "b" from the figure, as we considered that this distinction was not essential for the understanding of the results. The terms "a" and "b" referred to the different statistical comparison groups used to indicate where significant differences occurred between the treatments.
Comments 8: L 185: The scientific name Hydrilla verticillata should be written full only the first time in the text. Please, correct it, and in the case of other species.
Please, change the term "global biomass" to "total biomass" in the whole manuscript.
Response 8: Thank you for the suggestion. The scientific name Hydrilla verticillata has been written in full only at the first mention, as indicated, and the subsequent occurrences are in accordance with nomenclature rules. We have also made the same correction for the other species cited in the manuscript.
Additionally, we have changed the term "global biomass" to "total biomass" throughout the manuscript to ensure precision and clarity in the terminology.
Sincerely,
Letícia da Costa
On behalf of all co-authors

Round 2
Reviewer 1 Report
Comments and Suggestions for Authors
The authors have done a great job in improving the article. After revising the Results section, the article has become much clearer. The Conclusions have been greatly improved. In my opinion, the article is now ready for publication in Plants.
Reviewer 3 Report
Comments and Suggestions for Authors
Where are different letters that represent significant differences in mean biomass in Figure 2, 3 and 4? Please, correct it.